# Risk Stratification to Guide Prevention and Control Strategies for Arboviruses Transmitted by *Aedes aegypti*

**DOI:** 10.3390/tropicalmed8070362

**Published:** 2023-07-14

**Authors:** Manuel Osvaldo Espinosa, Verónica Andreo, Gladys Paredes, Carlos Leaplaza, Viviana Heredia, María Victoria Periago, Marcelo Abril

**Affiliations:** 1Fundación Mundo Sano, Paraguay 1535, Buenos Aires C1061ABC, Argentina; 2Instituto de Altos Estudios Espaciales Mario Gulich, UNC-CONAE, Falda del Cañete, Córdoba X5187XAC, Argentina; 3Consejo Nacional de Investigaciones Científicas y Técnicas (CONICET), Godoy Cruz 2290, Buenos Aires C1425FQB, Argentina; 4Hospital Juan Domingo Perón, Alberdi 855, Tartagal A4560AQI, Argentina

**Keywords:** risk stratification, hotspots, dengue, *Aedes aegypti*, Argentina

## Abstract

Strategies for the prevention of arboviral diseases transmitted by *Aedes aegypti* have traditionally focused on vector control. This remains the same to this day, despite a lack of documented evidence on its efficacy due to a lack of coverage and sustainability. The continuous growth of urban areas and generally unplanned urbanization, which favor the presence of *Ae. aegypti*, demand resources, both material and human, as well as logistics to effectively lower the population’s risk of infection. These considerations have motivated the development of tools to identify areas with a recurrent concentration of arboviral cases during an outbreak to be able to prioritize preventive actions and optimize available resources. This study explores the existence of spatial patterns of dengue incidence in the locality of Tartagal, in northeastern Argentina, during the outbreaks that occurred between 2010 and 2020. Approximately half (50.8%) of the cases recorded during this period were concentrated in 35.9% of the urban area. Additionally, an important overlap was found between hotspot areas of dengue and chikungunya (Kendall’s W = 0.92; *p*-value < 0.001) during the 2016 outbreak. Moreover, 65.9% of the cases recorded in 2022 were geolocalized within the hotspot areas detected between 2010 and 2020. These results can be used to generate a risk map to implement timely preventive control strategies that prioritize these areas to reduce their vulnerability while optimizing the available resources and increasing the scope of action.

## 1. Introduction

Arboviruses transmitted by the *Aedes aegypti* mosquito are a growing public health problem in countries in tropical, subtropical, and temperate areas of the planet. In the Americas alone, 500 million people live in areas with the presence of this vector and other Aedes species with vectorial capacity and proven circulation of Dengue, Chikungunya and Zika viruses [1]. Several factors have modulated the emergence and re-emergence of these diseases in the region [2,3]. Climate and atmospheric phenomena such as El Niño and La Niña impact on vector populations and expand the geographical distribution of the disease. They also extend the transmission period, impacting on morbidity and mortality rates. Continual migration from rural areas to urban centers has led to processes of disorderly urbanization and poor planning, generating scenarios that favor high densities of *Ae. aegypti* and viral transmission processes [2,3]. The lack of specific treatment drugs and available vaccines have also contributed to this expansion [1,2]. Although several vaccine candidates potentially effective against all four dengue virus serotypes have been developed over the past 20 years, they are not yet ready for use in public health.

Historically, initiatives to reduce the incidence of dengue in the Americas have focused on mosquito vector control [2,3,4]. The geographical dispersion observed and the recent introduction of chikungunya and Zika viruses highlight the limitations of control programs to effectively contain their spread. In recent years, this strategy has been revised due to the limited evidence as to its effectiveness, coverage, and sustainability [5,6]. The limited response capacity at local level to sustainably maintain this strategy is often very limited and is strengthened almost exclusively in epidemic times. The main drawback of this approach is the continuous growth of urban areas, together with the complexity of the urbanization phenomena that are generally not planned. The synergy of these factors generates scenarios with high population density and a large displacement of people [7]. In these scenarios and favored by the climatic conditions of the summer season, high levels of infestation of *Ae. aegypti* occur that reach high risk values [2,3]. Changes in the levels of immunological susceptibility of the inhabitants, a product of natural population turnover and migrations from rural areas, are essential components for the transmission of dengue and other arboviruses transmitted by this mosquito [8]. Consequently, efforts to achieve greater coverage, surveillance and control activities require greater resources, both material and human, which are not always feasible in most countries affected by these diseases [3]. On the other hand, no clear evidence has been documented on the effectiveness and sustainability of control efforts, especially when it comes to urban areas where the coverage of actions constitutes a critical point of the implemented programs [5,6].

Mundo Sano Foundation has been implementing a public–private collaboration program for the surveillance and control of *Ae. aegypti* since 2009 in the city of Tartagal, in northwestern Argentina. This program began after the epidemic of 2009, when Tartagal registered 1300 cases in the first 19 weeks of the year, which represented 48.5% of the cases in the province of Salta, and 2 deaths [9]. The program is aligned with the vector control strategies recommended by the PAHO [2]. It has focused on neighborhood inspection for the survey of infestation rates, monitoring of oviposition activity, and control strategies to reduce the urban abundance of breeding sites. The activities carried out have been planned according to the epidemiological periods. During the inter-epidemic period, activities include physical elimination and destruction of potential breeding sites. During the epidemic period, when a case of arbovirus is suspected—reported by the local Hospital—control with the BTI (*Bacillus thuringiensis* subspecies *israelensis*) biolarvicide and focal interruption of transmission with adulticides is implemented [9]. Although the program has achieved high adherence from the municipality and the community since the beginning, the greatest difficulty has been to achieve the desired coverage of the entire urban area, mainly in epidemic periods when activities and resources are concentrated on outbreak control.

In the search for alternatives to optimize the allocation of available resources, stratification analysis to identify risk areas, with recurrent transmission and high concentrations of the number of cases during dengue outbreaks, is a strategy of great operational utility [3]. These areas could be prioritized for the implementation of prevention and control actions [3,10,11]. Based on this approach, dengue cases registered in Tartagal in the period 2010–2020 were grouped by census tracts, with the aim of identifying high-risk areas to prioritize and to plan prevention and control activities, following the methodology proposed by the Pan American Health Organization/World Health Organization (PAHO/WHO) [3].

## 2. Materials and Methods

### 2.1. Study Area

The city of Tartagal [22°31′ S, 63°48′ W] is located in northwestern Argentina, 100 km north of the Tropic of Capricorn and 55 km south of the border with Bolivia (Figure 1). The urban area covers 21.3 km^2^. In 2010, the census population was 69,225 inhabitants [12] and in 2017 it was updated to 89,916 inhabitants, which meant an increase of 30% [13]. The geographic database of the city, prepared by the National Institute of Statistics and Censuses (INDEC), divides the surface of the city into 58 census tracts (CT) that have an average area of 0.37 km^2^ and a housing and population density of 1557 dwellings/km^2^ and 5740 inhabitants/km^2^ [13]. The region is characterized by a warm subtropical climate with an average annual temperature (2009–2020) of 22 °C and a dry season between May and October. December and January are the warmest months with an average temperature of 27 °C and July is the coldest with an average temperature of 15 °C. The average annual rainfall was 1025.7 mm during the period 2009 to 2020, with 2010 being the driest year with 678.8 mm and 2013 the wettest year with 1561.06 mm [14]. The Tartagal River crosses the urban area from west to east, dividing the city into north and south, while National Route 34 crosses it longitudinally, which constitutes a permanent transit of transport and people daily (Figure 1) and, consequently, an important source of virus in periods of viral circulation.

### 2.2. Dengue Cases

The dengue cases reported during the study period (2010–2020) were provided by the “J. D. Perón” Provincial Hospital of the city of Tartagal. This hospital is the reference center for the General San Martín Department of the province of Salta. The cases documented by the hospital correspond to patients who were seen daily during outbreak periods in each year, classified as Nonspecific Acute Febrile Syndrome and reported for focus control interventions. The cases included in this study correspond to those registered in the urban area of the city and that were confirmed by laboratory and/or epidemiological link, according to the algorithm indicated by the national health system [15]. The evolution in the number of dengue cases during the study period includes as a reference the number of cases reported during the 2009 outbreak. To explore the temporal pattern of the epidemic period, a stacked bar graph was used with the number of cases per week (Appendix A: Weekly distribution of dengue cases in Tartagal, Salta, Argentina from 2010 to 2020). The inclusion criterion for spatial analysis was to have the geolocation data of the cases. During 2016, together with the dengue outbreak, the first cases of chikungunya were registered in Argentina, which enabled the analysis of the degree of overlap of the hot spots identified for both diseases in Tartagal.

### 2.3. Data Analysis

Dengue cases were grouped by census tract (TC) for each outbreak year. The hotspot maps were created using QGIS 3.16 Hannover [16] and the Getis-Ord [Gi*] statistical model [17], which was run using 100,000 Monte Carlo simulations. This model is available in the HotSpotAnalysis add-on [18]. The Local Gi* spatial analysis method allows the identification of atypical locations in the spatial distribution of cases. The algorithm calculates a z-score for the clustered cases of each TC, which is then contrasted in a simple significance test based on the normal distribution. The null hypothesis indicates that the spatial distribution of events occurs completely randomly. The existence of groups or clusters with atypically high or low z-score values, with their respective probability values (*p*-value), would deny the existence of this null hypothesis. In other words, it confirms that there is no spatial randomness, but that the elements can be grouped spatially from the statistical point of view with their corresponding confidence intervals [17]. Hotspot maps were created for individual years, for 5-year periods (2010–2015 and 2016–2020) and for the entire period (2010–2020). Kendall’s W correlation coefficient (W = 0 without overlap; W = 1 perfect overlap) in the RStudio 2022.07.0 environment [19] was used to determine the degree of spatial overlap between dengue and chikungunya hotspots for the 2016 outbreak. Cases reported during 2022 were compared with hotspot areas for the entire period (2010–2020) to assess the degree of overlap and potential prediction.

## 3. Results

### 3.1. Temporal Patterns: Epidemic Period

Between 2010 and 2020, 9 dengue outbreaks were recorded in Tartagal with a total of 1270 cases (mean: 115.5, SD: 153.9); no cases were reported in 2010 and 2015 (Figure 2A). All outbreaks occurred in the first 25 weeks of each year, thus defining an epidemic period between the months of January to May and an inter-epidemic period from June to December (Figure 2B). The longest outbreak occurred in 2016 and lasted 21 weeks (weeks 1–21) with a peak of cases during weeks 8 and 9 (46 cases). The shortest outbreak occurred in 2017 with a duration of 12 weeks (weeks 9–20) and the peak of cases at week 16 (10 cases) (Appendix A: Weekly distribution of dengue cases in Tartagal, Salta, Argentina from 2010 to 2020). It should be noted that the epidemic period coincides with the rainy season of the year, a factor that alters the daily routine of carrying out vector monitoring and control activities.

### 3.2. Spatial Patterns: Hotspot Areas

Only 1012 of the reported cases (79.7%) fulfilled the geolocalization requirements for inclusion in the spatial analysis and the resulting yearly hotspot maps are shown in Figure 3. During 2011, 2014 and 2020, the areas with a significant Getis-Ord index were located in the central and northwest sectors of the city. During 2016, 2017, 2018 and 2019, hotspot areas were in the eastern sector of the city. Meanwhile, in 2012 and 2013, both previously mentioned hotspot areas appeared as critical (Figure 3).

Recurrence in significant critical areas was observed between 2010 and 2020 with TCs that were hotspots for up to four years (Figure 4A) and two TCs that constituted a hot spot for three consecutive years (Figure 4B). The stratification analysis carried out by five-year periods showed that for the period 2010–2015, the critical area was located in the northwest sector of the city (Figure 4C), while in the period 2016–2020, it was located in the east-central sector (Figure 4D). The global analysis (2010–2020) identified two urban clusters made up of 19 TCs that represent 35.9% of the total urban area and in which 51.0% of the accumulated cases throughout the period were concentrated (Figure 4E). These sectors of the city showed a high dynamism of urbanization during the period 2010 to 2020. 

In the summer of 2016, along with the dengue outbreak in which 82 cases were recorded, local transmission of the chikungunya virus was verified in Tartagal. A total of 139 cases of chikungunya were confirmed, including 8 cases with dengue coinfection (Appendix A: Weekly evolution of dengue and chikungunya cases in Tartagal, Salta, Argentina during the 2016 outbreak). Spatial transmission patterns for both viruses were analyzed and a high level of overlap was observed between the hotspot CTs identified for both diseases (Kendall’s W = 0.92; *p*-value < 0.001). We identified 11 TCs hotspots for chikungunya, seven of them overlapping with the TCs identified for dengue (Appendix A: Overlapping of dengue and chikungunya hotspots). During 2022, 55 cases of dengue were reported and 65.9% of them were located within the previously identified focus areas (Figure 5). 

## 4. Discussion

The effectiveness of efforts to maintain low levels of *Ae. aegypti* infestation and sustain them over time in ever-growing urban areas is unclear. The identification of high-risk scenarios for dengue is a necessary strategy to incorporate into dengue-control programs. The importance of prioritizing risk areas to plan strategies and schedules has been documented, seeking to optimize resources and personnel to achieve greater coverage and impact from the actions carried out [3,10,11,20,21]. The spatial distribution of *Ae. aegypti* in an urban area is heterogeneous, with highly focused populations depending on the abundance of breeding sites and is closely associated with human activity [3,11,22]. Consequently, at the beginning of a dengue epidemic outbreak, a similar phenomenon could be expected to occur with the spatial distribution of cases, with a few urban patches concentrating a high number of cases [10,11]. The results presented here show that all outbreaks that occurred between 2010 and 2020 in the city of Tartagal were concentrated in the urban area north of the Tartagal River, constituting recurrent foci in the 11 years. Two main areas located on the periphery of the main urban area were identified, one of them in the northwest and the other in the east-central area, in which 51% of the dengue cases documented between 2010 and 2020 were concentrated. The population living in these two conglomerates was 32 thousand inhabitants in 2017, which represented 46.22% of the population of that year [13].

The behavior shown by the dengue outbreaks that occurred in Tartagal between 2010 and 2020 reflect the condition of epidemiological vulnerability that characterizes the neighborhoods that make up the city and that are accentuated in peripheral areas. The main determinants documented during these years were the high abundance of potential breeding sites found in the houses. Furthermore, the presence of water tanks due to problems of access to drinking water increases in the summer–autumn seasons. These factors lead to high levels of mosquito infestation at a time of year that coincides with the period of epidemiological risk in the region [9]. The National Housing Pre-Census carried out in 2020, reported that between 2010 and 2020 the dengue hot spot areas grew from 14,915 to 21,793 homes, respectively, which represents an increase of 46.11% in 11 years [23]. The increase in the number of dwellings implies a greater number of breeding sites and levels of infestation of *Ae. aegypti* increase with the number of houses and degree of urbanization (7). A study carried out in 2013–2014 within the framework of the Strategic Plan for Local Development of the Municipality of Tartagal—Salta [24], showed that neighborhoods located in the sectors, and which constituted hotspots between 2010 and 2020, had the lowest environmental quality index (Appendix A). The factors analyzed to measure this indicator were the distribution of the urban water network, the quality of housing construction, the access of households and the population to drinking water, the presence of toilets, and connection to the sewage network. These factors mainly reflect the disorderly nature of urban growth with poor planning and suboptimal conditions of infrastructure and public services. Of the settlements, 25% do not have an urban layout and the houses were built irregularly by the population on public or private land, 30% have a precarious connection to the drinking water network and 40% are not connected to the sewage network [24]. Another aspect that contributes to the health vulnerability observed in these critical sectors is the high concentration of individuals in small areas. Thirty percent of the inhabitants reported living in lots that grouped two or three families in annexed boxes sharing a single bathroom and water source [24]. A distinctive and epidemiologically important feature is made up of National Route 34 and Provincial Route 86. Both roads converge in the eastern sector of the city (Figure 1) and are traveled daily by a very high percentage of heavy traffic and people. They constitute an important link with the bordering countries of Bolivia, Paraguay, and the Argentine littoral region. This factor is associated with the displacement of individuals who, in periods of viral circulation, constitute a potential source of virus (7).

As a result of the unplanned urban expansion observed in the eastern and northwestern sectors of Tartagal, it is plausible to think of the existence of greater epidemiological vulnerability that favors dengue transmission processes. Our results detected hotspot areas in these sectors of the city and could support this hypothesis. In addition, migratory processes imply a greater number of susceptible individuals that, together with asymptomatic cases not detected by the health system, constitute the main source of virus for mosquitoes, increasing the probability of viral transmission in small areas with high concentrations of people, incurring a greater number of cases in a short time [22,25].

The city of Tartagal recorded the circulation of chikungunya in 2016. A high degree of overlap was found between the TCs identified as dengue and chikungunya foci (Kendall’s W = 0.92, *p* < 0.001). From the spatial point of view, the seven TC hotspots that overlapped with the TC dengue hotspots were contiguous with each other, covering an area of 2.28 km^2^. Similar levels of overlap were documented between areas affected by concomitant outbreaks of dengue, chikungunya and Zika in eight cities in Mexico during the 2015 and 2016 epidemics [10,11]. Since both viruses are transmitted by the same mosquito species, the transmission processes would be determined by the same environmental and sociocultural factors. In areas with high infestation of active female mosquitoes and high circulation of susceptible and viremic individuals, overlapping spatial patterns of hot spots would be observed during the cocirculation of *Ae. aegypti*-borne viruses [2,10,11].

In Argentina, *Ae. aegypti* has shown markedly seasonal dynamics. Its peak of activity and infestation has been recorded between the months of December and April, coinciding with the period of greatest probability of viral circulation in the region [26]. The adaptive capacity of *Ae. aegypti* has been reflected in recent outbreaks in Argentina, characterized by an increase in the affected area and increased reporting of cases, especially in regions that had never reported cases of local transmission [27]. The gradual increase in temperature recorded in recent years in our region would be associated with the increase in the geographical areas affected and a longer duration of epidemic periods [27]. Tartagal’s documented outbreaks occurred during the first 25 weeks of the year, with a peak of cases between weeks 11 and 17. The shortest outbreak occurred in 2011 (10 weeks) and the most extensive outbreak was in 2020 (18 weeks) (Appendix A). The climatic trend previously mentioned could influence the temporal pattern observed in Tartagal, extending the epidemic period of viral circulation, and enhancing transmission processes.

In the 2022 dengue outbreak, 61% of confirmed cases were located in the critical areas identified for the period 2010–2020. These results allowed us to verify the usefulness of the stratification methodology in the recognition of potential risk scenarios and to generate crucial information to prioritize preventive activities during the following inter-epidemic period. However, the dynamic nature of the spatial patterns observed in each of the outbreaks that occurred between 2010 and 2020 reinforces the need to repeat the risk stratification analysis at the end of each epidemic period to detect the presence of possible foci in sectors of the city that were not shown as critical areas in previous outbreaks.

The program implemented in Tartagal since 2009 consists of an operational schedule of activities based strongly on the local epidemiological behavior of dengue and the seasonal dynamics of the mosquito [9]. The temporal pattern of dengue outbreaks documented during the decade 2010–2020 supports the temporal distribution of preventive actions carried out during the year (Appendix A: Operational scheme for the activities developed throughout the year as part of the Surveillance and Control Program for Aedes aegypti implemented in the city of Tartagal, Salta, Argentina). The stratification analysis described here will greatly contribute to the prioritization and strengthening of the strategies implemented in the areas identified as critical. These strategies should be reinforced especially in early warning periods, in the weeks prior to the start of a potential period of high transmission.

The main limitation of this study is obtaining the geolocation data of the total number of patients. In many cases, patients did not report their real address or indicate references that were not later located, a factor that added to the difficulty in finding neighborhoods in growing urbanizations and with little planning in peri-urban and rural areas. Moreover, the residence was not necessarily the place where the person became infected [28]. However, the house is a place of high permanence for an infected individual, potentially in viremia, therefore a place of increased risk to act as a focus of transmission for the block and the neighborhood. In this regard, it has been documented that the spatial application of adulticides in places with high exposure potential, such as the home of a confirmed case, reduces the likelihood of future dengue transmission by up to 96% compared to places without treatment [29]. Since only cases detected by the health system were considered, the actual number of people affected tended to be underestimated [22,25]. Furthermore, the mild or subclinical infections generally do not require medical attention, so when considering the movement of people within an urban area, these individuals constitute a source of dispersion and intensity of transmission in an epidemic outbreak [30].

## 5. Conclusions

The main characteristic of arboviral diseases transmitted by *Ae. aegypti* in Argentina is the epidemic and seasonal nature of the outbreaks. An advantage of seasonality is the possibility of implementing control actions and intensifying preventive strategies during inter-epidemic periods and low levels of *Ae. aegypti* infestation. 

Due to the limited coverage of activities aimed at vector control, the detection of areas that present unusual behavior during a dengue outbreak allows the prioritization of these sectors in the planning of activities. In addition, it allows us to focus on these hotspot areas to carry out studies aimed at recognizing the determining factors that play a preponderant role in the onset and subsequent evolution of an epidemic outbreak. This information is indispensable to design strategies specifically aimed at the determinants of abundance and infestation.

The stratification analysis is presented as an ideal operational tool for the identification of the strata of the city that would present the greatest risk of transmission in periods of viral circulation. This approach is part of the guidelines proposed by PAHO/WHO to achieve a focused and rational management of this mosquito in urban areas, with the aim of achieving a greater impact and optimizing the allocation of resources in space and time [3]. The next step will be to measure the impact of preventive interventions focused on vulnerable areas or hotspots and generate evidence that contributes to developing an approach model that can be extrapolated to other municipalities, within the framework of strengthening efforts to mitigate the impact of arboviruses transmitted by *Ae. aegypti* in the region. 

## Figures and Tables

**Figure 1 tropicalmed-08-00362-f001:**
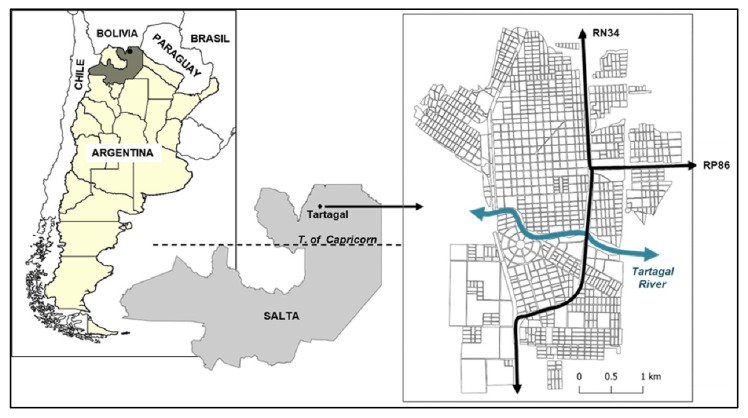
Geographic location of the city of Tartagal, Salta province (Argentina) with respect to the region and the province on the **left** and the urban map of the city on the **right**.

**Figure 2 tropicalmed-08-00362-f002:**
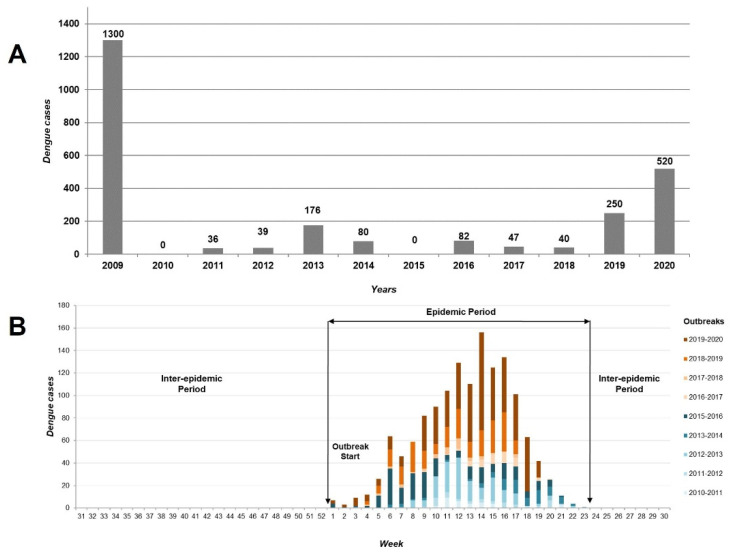
Number of dengue cases per year during the study period (2010–2020) in Tartagal, Salta (Argentina) (**A**) and the cumulative epidemic curve showing the registered epidemic curve during the same period (**B**).

**Figure 3 tropicalmed-08-00362-f003:**
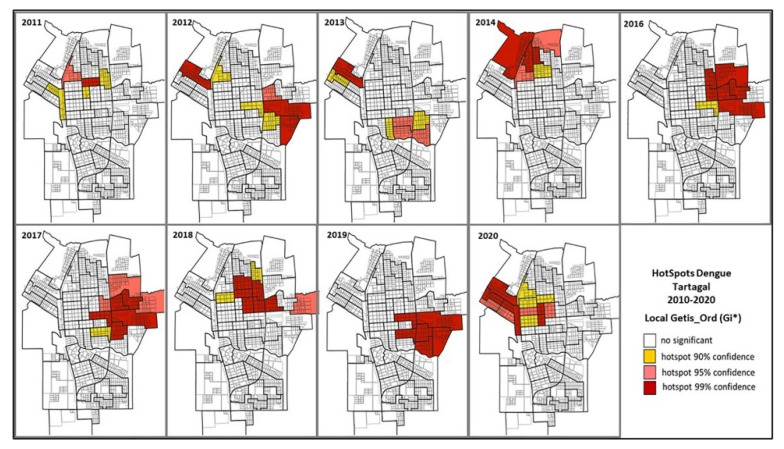
Census tracts (CTs) in the city of Tartagal, Salta (Argentina) and the yearly dengue hotspots obtained using Getis-Ord indexes, 2010 to 2020. Maps for the years 2010 and 2015 are not included given the lack of outbreaks.

**Figure 4 tropicalmed-08-00362-f004:**
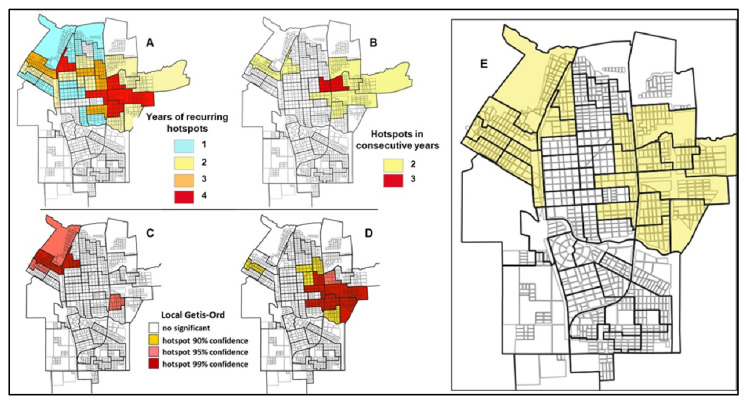
Evolution of hotspots throughout the study period (2010–2020) in Tartagal, Salta (Argentina). (**A**) Recurrent dengue hotspots per census tract (CT), (**B**) number of consecutive years that the same hotspot was identified as such, (**C**) dengue hotspot areas detected from 2010 to 2015, (**D**) dengue hotspot areas detected from 2016 to 2020, and (**E**) accumulated dengue hotspots per CT during the entire study period.

**Figure 5 tropicalmed-08-00362-f005:**
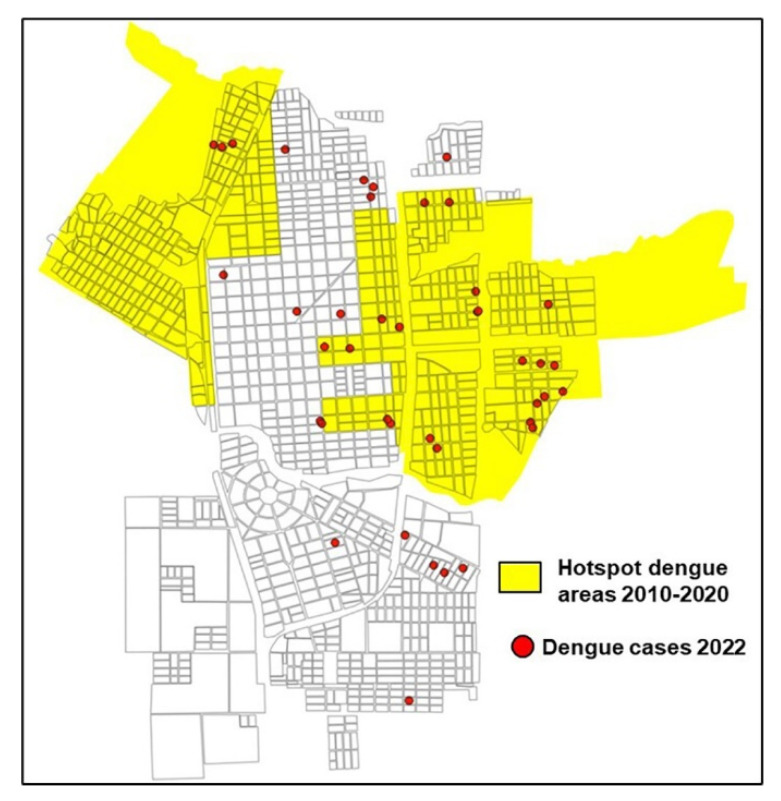
Mapped distribution of confirmed dengue cases during the 2022 outbreak for Tartagal, Salta (Argentina) with respect to the hotspot areas detected during the period 2010–2020.

## Data Availability

Not applicable.

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
