# Peer review of "Risk Stratification to Guide Prevention and Control Strategies for Arboviruses Transmitted by Aedes aegypti"

_tropicalmed, 2023, doi:10.3390/tropicalmed8070362_

Round 1

Reviewer 1 Report

The strategic and methodological design is excellent, and the results are precise.

Minor corrections

Figure 2-4 increase the resolution

Line 46, 226 “Ae. aegypti

Line 228 is missing references in the text

What local and national strategies could be implemented in Argentina to register all cases of flaviviral infections and thus implement your model in your nation? And how to extrapolate it to other countries?

Reviewer 2 Report

This manuscript describes an interesting idea and deals with vector borne diseases and viruses transmitted by Aedes aegypti mosquitoes such as Dengue, Chikungunya and Zika and the risk stratification to guide prevention and control strategies for viral transmission.  

First of all, I believe that in section 2.2 the authors should provide more information regarding cases and their recording. Is it through a national/ local surveillance epidemiological tool? Is it through a system of mandatory recording of human cases? In section 2.2 the authors state that these are cases provided by the provincial hospital. What percentage of cases do they represent? Are there other hospitals or medical/community centers in the area? This is an important point as the authors use this data as the basis of their analysis.

I also believe that this manuscript could benefit from the presentation in the Results section of the two figures that are now supplied as supplementary figures, figure 1A and 1B regarding the CHIK virus cases and the common hotspot Dengue and Chikungunya areas.

I have a few suggection for the discussion / conclusion sections. My opinion is that the conclusions of the authors are based on findings that cannot be considered as final, given that the data used was not exhaustive (not sure that all human cases were recorded, only data from local hospital, under-recording possible?) and also as they describe in the limitations’ paragraph, the authors cannot be sure about the exact areas where the patients got infected. This data is of importance for analysis and for supporting conclusions. More data is needed to support major changes and new approaches. Future strategies regarding prevention and control  of diseases transmitted by Aedes aegypti mosquitoes should not only be focused on hotspot areas and change the general approach, reduce interventions in other areas, as it is known that it is not easy to predict with accuracy from one year to the other, areas and places where diseases transmitted by mosquitoes will be detected.  Therefore, it is essential for all countries / Regions/ municipalities to make an effort on maintaining their national/local control strategies and interventions on an annual basis and then work as the authors on identifying hotspots and vulnerable areas early enough, propose and then apply extra interventions in the identified areas as hotspots. To my opinion, data presented here can support this type of conclusions and suggestions and not a general strategy change. I encourage the authors to follow this approach in the discussion and conlcusion, as they succesfully do in the last sentence of the abstract. .

Minor

-Throughout the text all mosquitoes’ names e.g. (Aedes aegypti) should be written in italic

-Line 46 replace with: public private collaboration program

-Lines 47- 48 replace with:  when Tartagal recorded 1,300 daily cases representing 48.5% of the total cases

- Line 196: check that references 33 to 40 describe this sentences

-In conlusions section, references are missing

Reviewer 3 Report

This manuscript describes the application of a risk stratification approach to identify dengue hotspots in a city of Argentina based on cases reported in the previous decade. It then maps more recent cases to these hotspots, and suggests that the hotspots could be updated every year to determine which parts of the city should be targeted for vector control prevention strategies.

The manuscript is well written, and the conclusions justified based on the author’s findings. The discussion is well structured and cites relevant articles. It recognises the importance of environmental factors in the hotspot areas and the need to revisit the models each year.

I have made minor recommendations below.

Line 34: Capitalise Zika

Line 36: Perhaps mention the lack of evidence for their effectiveness ((e.g. See Bowen et al., PLoS Negl Trop Dis. 2016 or something more recent).

Line 50: Can you explain what is meant by environmental burden of breeding sites – do you mean reduce the number of them?

Line 69: It looks more north-western in the figure.

Line 94: Please explain what Getis-Ord index is, for those not familiar.

Line 108: Clarify what is meant here – there are more than 21 weeks between week 1 and week 26.

Figures 2 -5 look a bit blurred. This might just be the reproduction in the proof but worth checking they are crisp in the final version.

Line 116: It isn’t clear to me what the requirements were – were come cases outside the city bounds? Please clarify in the methods section why some would be excluded.

Line 128: Would it be interesting to have stratification of a rolling 5-year period? E.g. years 2010-2014, 2011-2015, 2012-2026 etc, rather than splitting the study period into two halves.

Figure 5 has recent dengue cases shown at quite a fine scale. Is there any risk of households being identified from this map?

Lines 157-159: I’m not sure this is correct. If you made a composite image of the maps shown in Figure 3 I think it would show that a large part of the northern half of the city had outbreaks, and Figure 5 shows a good proportion of the city was a hotspot in the period 2010-2020.

Line 181: Add ‘species’ - the same mosquito species.

Line 200: Perhaps suggest here that changes in climate may affect the timing or duration of epidemics.

Line 228: Please find suitable references to add here, and give the protocol another read through to check for minor corrections (e.g. italicisation on line 226).

Round 2

Reviewer 1 Report

The article meets the requirements for publication

Reviewer 2 Report

The points I raised, especially in the Discussion / Conclusion section have been addressed by the authors. The manuscript has been substantially improved.

Reviewer 3 Report

Thank you for carefully addressing the comments made in my first review.